# Normative Data of Supraspinatus Muscle Shear Wave Elastography in Healthy Shoulders: A Cross-Sectional Study

**DOI:** 10.3390/jcm14041121

**Published:** 2025-02-09

**Authors:** Irene Pérez-Porta, Ángel Luis Bueno-Horcajadas, Fernando García-Pérez, Diana Cecily Martínez-Ponce, Silvia Corrales-Mantecón, Mariano Tomás Flórez-García, María Velasco-Arribas

**Affiliations:** 1Physical Therapy and Rehabilitation Unit, Hospital Universitario Fundación Alcorcón, 28922 Alcorcon, Madrid, Spain; fernando.garcia.perez.1961@gmail.com (F.G.-P.); scorralesm2022@gmail.com (S.C.-M.); marianotomasflorez@gmail.com (M.T.F.-G.); 2International Doctoral School, Universidad Rey Juan Carlos, 28933 Mostoles, Madrid, Spain; 3Diagnostic and Interventional Musculoskeletal Radiology Unit, Hospital Universitario Fundación Alcorcón, 28922 Alcorcon, Madrid, Spain; angelluis.bueno@salud.madrid.org; 4Occupational Risk Prevention Service, Hospital Universitario Fundación Alcorcón, 28922 Alcorcon, Madrid, Spain; dianacecily.martinez@salud.madrid.org; 5Research Unit, Hospital Universitario Fundación Alcorcón, 28922 Alcorcon, Madrid, Spain; mvelascoa@salud.madrid.org; 6Department of Medical Specialties and Public Health, Universidad Rey Juan Carlos, 28933 Mostoles, Madrid, Spain

**Keywords:** shear wave elastography, shoulder, normative, ultrasound imaging

## Abstract

**Background/Objectives**: In the shoulder region, shear wave elastography (SWE) has been used to obtain data from multiple muscles. However, there is still a lack of evidence regarding normative values for the supraspinatus muscle. The aim of this study is to estimate the range of normative values and to evaluate the relationship between SWE measurements and isometric strength. **Methods**: A cross-sectional study with 46 healthy subjects was conducted. Data regarding the SWE of supraspinatus muscle at rest and during contraction and isometric elevation strength were collected. Ordinal cumulative probability models were implemented to calculate normative values based on age and sex. **Results**: There was a significant increase in muscle stiffness from rest to contraction (3.97; 95% CI, 3.52 to 4.43), but there were no differences between males and females. The ordinal regression models showed a relationship between age and SWE at rest (coefficient, 0.08; 95% CI, 0.01 to 0.14), but not during contraction, and there was no significant age–sex interaction. Normative values of the median and 25th and 75th percentiles were provided based on individuals’ age and sex. There was no correlation between SWE measurements and strength values. **Conclusions**: Normative values for supraspinatus muscle SWE measurements at rest and during contraction were obtained. These data can help clinicians to interpret measurements of their patients with shoulder disorders.

## 1. Introduction

The shoulder girdle is composed of the clavicle, humerus, and scapular bones, which work in a coordinated manner to position the arm in a specific three-dimensional location in space. For this purpose, there are up to 17 muscles that induce movements at the level of the glenohumeral, acromioclavicular, and sternoclavicular joints [1]. In the glenohumeral joint, there is a group of muscles of particular importance for arm movements; they provide both torque and stability to this region, known as the rotator cuff. These include the supraspinatus, infraspinatus, teres minor, and subscapularis, which have optimal lines of action to provide stability at the glenohumeral level and to produce powerful torque for movement in various directions [2].

The fact that this musculature is so heavily involved in all arm movements means it endures considerable stress over a lifetime, making degenerative tears in its tendons common, both in people with shoulder pain and in asymptomatic individuals [3]. For this reason, although there does not appear to be a strong relationship between tendon tears and the onset of shoulder symptoms [3], many researchers have focused their investigations on studying the function of these muscles [2,4] and their possible dysfunction in patients with shoulder complains [5]. Among the rotator cuff muscles, the supraspinatus is the most exposed to mechanical stress due to its role as an abductor, flexor, internal rotator, external rotator, and stabilizer [2]. This situation makes the musculotendinous complex of the supraspinatus the most involved in rotator cuff injuries, both traumatic and atraumatic, making it a good reference point for obtaining values that reflect the overall functional state of the rotator cuff [6,7].

The analysis of muscle function has been a topic of interest for decades, and it has traditionally been measured using a variety of tools, including electromyographic signals, dynamometers, isokinetic devices, and ultrasound imaging for muscle thickness [8,9].

In recent years, there has been an increase in interest in another method for evaluating muscle function that the previous tools could not measure; this method is shear wave elastography (SWE). This technique involves inducing shear waves through an external force, where the speed of wave propagation through tissues allows us to estimate tissue elasticity, such as the elasticity of a muscle belly at rest and during contraction [10]; previous research has shown a correlation between SWE and muscle torque and/or electromyographic signals [11].

In the shoulder region, SWE has been used to obtain data in multiple muscles, such as the trapezius, levator scapulae, rhomboid, deltoid [12], and rotator cuff muscles [12,13,14,15]. Among these, the supraspinatus muscle is the most commonly measured [16]. Previous published studies have investigated the intra- and inter-rater reliability of SWE measurements [17], compared subjects with and without shoulder pain [18], or analyzed the effects of therapeutic exercise programs or surgery on SWE measurements [19,20]. On the other hand, there is only a small amount of published literature analyzing normative values of SWE measurements in the supraspinatus muscle belly [13,21,22]. This information is crucial because it facilitates the interpretation of SWE measurements of patients with shoulder disorders in clinical practice, which could help to improve the diagnosis of shoulder disorders, enhance the monitoring of the clinical progression of patients, facilitate the implementation of individualized treatment selection, and allow the early detection of muscle tissue alterations, with the aim of improving risk prediction to avoid possible musculoskeletal injuries [21]. However, there are not enough data to accurately estimate the range of normative values, and most of the studies conducted have flaws in their statistical analyses, making it difficult to draw definitive conclusions on this topic [13,21,22].

The aim of this cross-sectional study is to analyze SWE measured in the supraspinatus muscle belly, at rest and during contraction in healthy shoulders, and to estimate a range of normative values. Furthermore, the relationship between SWE and hand-held dynamometer measurements is also analyzed.

## 2. Materials and Methods

### 2.1. Design

This was a cross-sectional study conducted as per the recommendations of Strengthening the Reporting of Observational Studies in Epidemiology (STROBE). The study was conducted following the Declaration of Helsinki and approved by the ethical committee of Hospital Universitario Fundación Alcorcón (reference number 20/151).

### 2.2. Setting

The study was conducted in Hospital Universitario Fundación Alcorcón (Madrid, Spain) from January 2021 to December 2024. All the participants signed an informed consent before any measurement procedure.

### 2.3. Participants

The participants were recruited from Hospital Universitario Fundación Alcorcón (Madrid, Spain). The healthy shoulders of patients referred for rotator cuff-related shoulder pain was included, as well as the dominant shoulders of hospital employees without any shoulder complaints.

To be able to participate in the study, subjects should fulfill the following inclusion criteria:Age between 18 and 80 years.Able to understand written and spoken Spanish language.

Furthermore, patients should not present with any of the following exclusion criteria:
History of major trauma or surgery on the shoulder, elbow, or cervical spine.Signs of other shoulder pathologies, such as instability, frozen shoulder, calcific tendonitis, severe arthrosis, or neuralgic amyotrophy.The presence of full-thickness rotator cuff tears on ultrasound imaging.Signs and/or symptoms of neck-related shoulder pain and/or radiculopathy or radicular pain.Systemic diseases such as cancer, rheumatic disorders, sclerosis multiple, neurological disorders, etc.Severe psychiatric disorders.


The sample size was set a priori based on previous literature [23,24,25]. It was established that a minimum of 40 subjects were needed to successfully obtain reliable data on the normative values. Since most of the supraspinatus disorders were presented by those aged between 40 and 60 years, we aimed to recruit more subjects within this range to estimate more reliable normative values. The minimum final sample was composed of 10 subjects between 20 and 40 years, 20 subjects between 41 and 60 years, and another 10 subjects above 60 years old.

### 2.4. Measurement Procedures

After the recruitment procedure, the following participant data were collected by the physicians: age, height, weight, sex, dominant side, side with pain, and time with pain. Data regarding SWE and isometric strength were measured by two radiologists who specialized in ultrasound imaging. One of the radiologists took the SWE measurements, while the other one helped with the positioning of the patient and with securing the hand-held dynamometer.

#### 2.4.1. Shear Wave Elastography

The SWE was measured using a Canon Aplio i600 (Canon Medical Systems Corporation, Tustin, CA, USA) ultrasound machine, with a linear probe (14 MHz) and a circular region of interest of 5 mm. The probe of the ultrasound was placed parallel to the long axis of the supraspinatus muscle belly, at its thickest area in the longitudinal plane. Each subject was measured three times at rest and during isometric contraction of elevation in the scapular plane, with a 30 s rest in between.

For the measurements taken at rest, the subject was seated upright in a chair, with feet flat on the floor and the arm positioned at 90° of elbow flexion, resting the forearm on the thigh. For the isometric maximum contraction measurements, the subject remained in the same position, but with the arm elevated to 75–80° in the scapular plane, exerting force against a strap placed on the lateral surface of the arm just above the humerus’s lateral epicondyle. This strap was attached to a digital dynamometer, and the subject held the contraction for five seconds [11,26].

For each measurement, the mean velocity (m/s) and standard deviation of the region of interest was calculated (Figure 1). The final value for each subject was calculated as the weighted mean of the three measurements, using the inverse of the standard deviation as the weights.

#### 2.4.2. Isometric Strength

Isometric strength was measured using a digital dynamometer (Carp Spirit Water Queen, Bourogne, France). The participants’ positioning was the same as previously described for the SWE measurements. Three measurements were taken, and the mean was used for statistical analyses.

### 2.5. Statistical Analysis

All statistical analyses were conducted using R software v.4.1.0 (R Core Team 2021. R: A language and environment for statistical computing. R Foundation for Statistical Computing, Vienna, Austria).

For the descriptive analysis of quantitative variables, the mean, median, standard deviation (SD), first and third quartiles, and range were reported. For categorical variables, the absolute frequencies and percentages were reported.

An intra-rater reliability analysis was performed with the SWE measurements at rest and during contraction, as well as with the isometric strength measurements. The root mean squared error (RMSE) and the minimum detectable change with 95% confidence bounds (MDC95%), calculated as RMSE * √2 * 1.96, were reported [27].

Normative data for SWE were calculated using an ordinal cumulative probability regression model with a logistic family, with predictors such as age, sex, and their interaction term. The normative values are presented as the fitted median and the 25th and 75th percentiles. The ordinal models were calculated using the function ‘orm()’ from the ‘rms’ (Frank Harrell, 2024) package in R.

Finally, the relationship between strength measurements and SWE is analyzed using an ordinary least squares regression model, with isometric strength as the dependent variable and the SWE measurements at rest and during contraction as the predictors of the model. The relationship is evaluated with the adjusted determination coefficient (adjusted R^2^), as well as with the unstandardized regression coefficients for SWE measurements at rest and during contraction.

All the analyses were conducted assuming an alpha value of 0.05, with 95% confidence intervals (CI).

## 3. Results

The final sample was composed of 46 subjects (32 female), with a mean age of 50.37 (SD, 13.16) years, and a mean BMI of 25.45 kg/m^2^ (SD, 4.36). The full summary of the descriptive statistics is presented in Table 1, and the descriptive statistics by sex are presented in Table 2.

### 3.1. Reliability Analyses

The RMSE for the SWE measurements at rest was 0.33 m/s, and during contraction, it was 0.94 m/s. For the strength measurements, the RMSE was 0.78 kg. The MDC95% for the SWE measurements was 0.93 m/s at rest and 2.62 m/s during contraction, and for the strength measurements, it had a value of 2.15 kg.

### 3.2. Changes in Shear Wave Elastography

There was an increase in the SWE measurements from rest to contraction (mean difference, 3.97; 95% CI, 3.52 to 4.43).

There were no differences between males and females in the SWE measurements at rest (mean difference, 0.40; 95% CI, −0.19 to 0.99), during contraction, (mean difference, 0.20; 95% CI, −0.57 to 0.96), nor were there any differences in the adjusted values during contraction (adjusted mean difference, 0.30; 95% CI, −0.45 to 1.04).

### 3.3. Normative Values of Shear Wave Elastography

The predicted normative values (median, 25th and 75th percentiles) of the SWE measurements of the supraspinatus muscle at rest are presented in Figure 2 and Table 3, and the values during contraction are presented in Figure 3 and Table 4. The pseudo-R^2^ value for the model at rest was 0.19, and during contraction, it had a value of 0.06. The model coefficients are presented in Table 5. There was a significant positive relationship between age and the SWE measurements at rest, but not during contraction. And there was a non-significant relationship of SWE at rest, but not during contraction with sex; nor was there a significant age–sex interaction (Table 5).

### 3.4. Relationship Between Shear Wave Elastography and Strength Measurements

There was no relationship between the SWE measurements and strength measurements (adjusted R^2^, −0.05). The data for the model coefficients are presented in Table 6.

## 4. Discussion

This was a cross-sectional study conducted in Hospital Universitario Fundación Alcorcón (Spain) that aimed to estimate the range of normative values of the SWE measurements of supraspinatus muscle belly at rest and during contraction, based on individuals’ age and sex, and to evaluate its relationship with strength measurements.

The results of the present study showed no significant differences between males and females in the SWE measurements of the supraspinatus muscle belly at rest (mean difference, 0.40; 95% CI, −0.19 to 0.99) or during maximum isometric contraction (mean difference, 0.20; 95% CI, −0.57 to 0.96). Some previous studies have found differences between males and females in the SWE measurements in some muscles, like the pectoralis major [28] or hamstring muscles [29], but there is also some research showing no differences in other muscles, like the gastrocnemius [30]. Furthermore, in studies conducted in the supraspinatus muscle, the evidence is contradictory, with some statistically significant differences found in some subregions but not in other ones, with no clear clinical relevance [22]. Overall, there is no definite evidence for differences between males and females in SWE measurements in muscle bellies. More research should be conducted on this topic to draw definitive conclusions.

On the other hand, there was a significant small positive relationship between age and SWE at rest based on the ordinal cumulative probability models (coefficient, 0.08; 95% CI, 0.01 to 0.14), but there was a non-significant negative correlation during contraction (coefficient, −0.01; 95% CI, −0.08 to 0.05). Some previous studies have found a similar positive association between age and SWE measurements in the supraspinatus muscle belly at rest and during muscle contraction (not maximum) [13,22]. There seems to be no clear explanation for the increase in muscle stiffness observed with advanced age at rest [13]. With advancing age there is a decrease in muscle thickness and an increase in the connective tissue (e.g., collagen) within the muscle bellies, which is more rigid than muscle fibers [31]. Furthermore, there can be changes in local muscle proteins that can lead to an increase in muscle stiffness at rest [32]. The fact that there was no significant relationship between age and SWE during maximum isometric contraction can be explained by the presence of a ceiling effect. The previous literature has shown a trend towards increased muscle stiffness with an increased percentage of maximum voluntary isometric contraction. It may be that, because we measured muscle stiffness at maximum voluntary isometric contraction, there is a ceiling effect that may not be present in the lower contraction intensities used in the previous literature [13,22]. Furthermore, despite not being significant, there seemed to be a negative relationship between age and SWE during contraction, which can reflect the decrease in the capability to produce muscle contraction with advancing age [32].

To the knowledge of the authors, there is only one published study which aimed to estimate the range of normative values of supraspinatus muscle belly stiffness [21]. In their study, Shimizu et al. [21] evaluated 43 participants between 24 and 75 years old, with a mean age of 46.7 (SD, 14.9). They measured stiffness by the strain ratio at 0° and 60° of shoulder abduction and calculated the range of normative values using normal distribution methods after applying logarithmic transformations. Furthermore, they categorized subjects into two subgroups based on age (<50 and ≥50 years old) [21]. Data categorization is not recommended, since it can lead to a wrong estimation of the association between variables and to spurious associations [33]. Furthermore, although they used a logarithmic transformation of the data, this procedure alone does not ensure that the statistical assumptions of parametric models are met. Finally, they did not take into account other covariates, such as sex, in the estimation of the normative reference values. Statistical methods, such as ordinal cumulative probability models, offer advantages in estimating the range of normative values and also in allowing the inclusion of multiple covariates [34].

In our study, we estimated the range of normative values of SWE measured in supraspinatus muscle belly at rest and during maximum isometric contractions, based on age and sex. Our data provide estimated median and 25th and 75th percentiles for each 1-year increase in age, separated by sex (Table 3 and Table 4). The ordinal cumulative probability model for SWE at rest showed a trend towards an interaction between age and sex; however, it was not significant. This may be because of the small sample size, since most interactions in regression analyses require greater sample sizes to be detected in hypothesis testing [35]. Our normative values can be used to discern whether a subject with a specific sex and age is far away from the values of healthy shoulders. Therefore, these data could be helpful for clinicians when making decisions with their patients, such as evaluating the effects of exercise programs [19] or surgical procedures [36]. However, there is a need for more studies with greater sample sizes to externally validate our estimated reference normative values.

Finally, we observed no relationship between the SWE measurements and isometric strength values (adjusted R^2^, −0.05). Previous literature has found an increase in SWE measurements with increased intensity during isometric contractions within a given subject [37,38]. However, when the contraction intensity is fixed (i.e., 40% of maximum voluntary isometric contraction) there seems to be no relationship between strength measurements and SWE measurements. Therefore, we can conclude that increasing contraction intensity leads to an increase in muscle stiffness, but we cannot predict muscle force based on SWE measurements at a given contraction intensity [37,38]. These results from the previous literature agree with the ones obtained in the present study.

In summary, the results of the present study can be used to discern whether a patient with shoulder symptoms is within the range of values of healthy subjects, implying that his muscle is within a normal range of mechanical properties. A subject who is well above or below the normative percentiles could have an underlying pathology (e.g., rotator cuff tendinopathy instead of subacromial bursitis) or be at higher risk of developing a future rotator cuff disorder (e.g., rotator cuff tear). Additionally, a subject outside the normative range limits may also respond poorly to a specific treatment (such as rotator cuff repair). Moreover, these data can be used to assess the patient’s progress after surgery or, for example, an exercise program to determine whether the muscle is responding adequately to the intervention.

The present study has some strengths and limitations. First, this is the first conducted study using advanced statistical methods for the estimation of the range of normative values based on multiple covariates, without categorizing continuous predictors. This allows precise and more valid estimations of normative values than the previous statistical methods implemented. One limitation of the present study is that we did not measure SWE in different subregions of the supraspinatus muscle belly. Furthermore, the small sample size limited the number of possible covariates to include in the regression models; so, we could not evaluate the influence of other factors, such as body mass index or physical activity levels. Finally, our study was conducted in a single center; so, our findings should be validated against other populations or datasets to improve applicability to diverse clinical settings.

Future studies should be conducted with greater sample sizes to externally validate our range of normative values and to allow the possibility of including more covariates (e.g., body mass index or physical activity level) and their interactions. Furthermore, future studies should be conducted using multicenter design and should include subjects with shoulder pathological conditions to improve the robustness and applicability of the results. Furthermore, future studies should also evaluate how these SWE measurements combine with other diagnostic tools and/or clinical assessments to improve the evaluation and management of shoulder disorders.

## 5. Conclusions

This cross-sectional study established normative median and 25th and 75th percentile values of SWE measurements in the supraspinatus muscle at rest and during contraction, by age and sex. However, the results should be interpreted cautiously due to the limited sample size, which restricts generalizability and may reduce the robustness of the obtained values. On the other hand, there seems to be no relationship between sex and SWE measurements, nor between SWE measurements and isometric strength measurements. More research should be conducted to externally validate the estimated range of normative values.

## Figures and Tables

**Figure 1 jcm-14-01121-f001:**
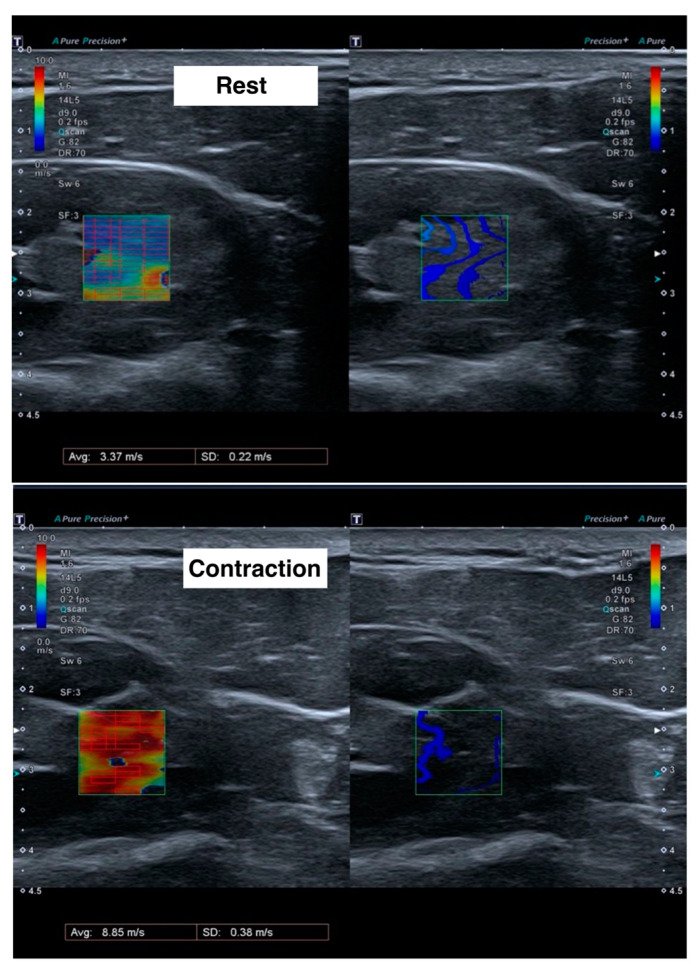
Ultrasound images during shear wave elastography measurements at rest (**upper**) and during contraction (**lower**). Left images show the velocity of the shear waves in meters per second, with red color meaning more speed and blue color less speed. Right images show the form of the shear wave.

**Figure 2 jcm-14-01121-f002:**
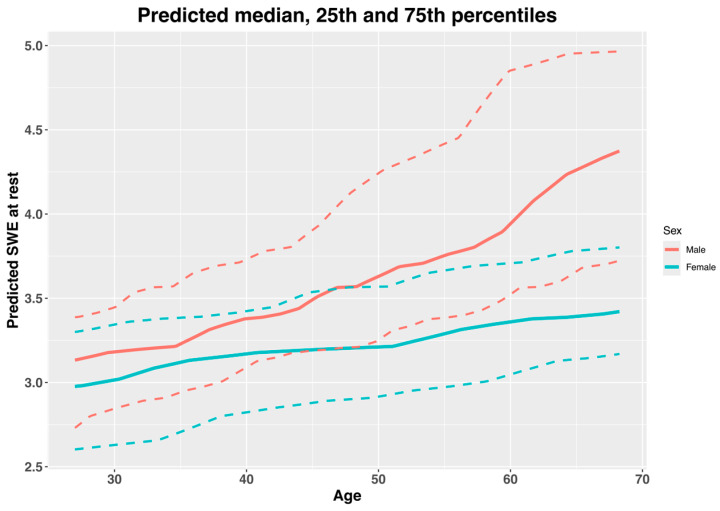
Predicted normative values of median, 25th and 75th percentiles of shear wave elastography (SWE) of supraspinatus muscle at rest, by age and sex. Solid lines are medians, and dashed lines are 25th and 75th percentiles. Data interpretation: There is an estimated increase in SWE measurements at rest with increasing age, and males show greater SWE values than females, especially in older ages.

**Figure 3 jcm-14-01121-f003:**
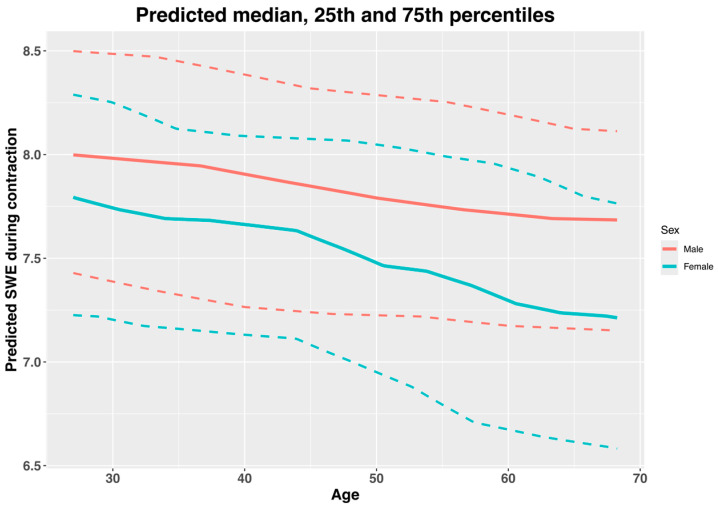
Predicted normative values of median, 25th and 75th percentiles of shear wave elastography (SWE) of supraspinatus muscle during contraction, by age and sex. Solid lines are medians, and dashed lines are 25th and 75th percentiles. Data interpretation: There is an estimated decrease in SWE measurements during contraction with increasing age, and males show greater SWE values than females, which do not appear to be influenced by age.

**Table 1 jcm-14-01121-t001:** Descriptive statistics (n = 46).

Variable	N	Mean	Std. Dev.	Min.	Pctl. 25	Median	Pctl. 75	Max.
Age, years	46	50.37	13.16	24	42	52.5	60	73
Height, cm	46	166.21	9.06	150	160	166.5	172.75	186
Weight, kg	46	70.60	14.83	45	62	69.5	77.75	119.4
BMI, kg/m^2^	46	25.45	4.36	18.49	22.05	24.53	28.15	38.20
Sex	46							
Male	14	30.4%						
Female	32	69.6%						
Education	46							
Primary school	8	17.4%						
Secondary school	10	21.7%						
University	28	60.9%						
Dominant side	46							
Right	43	93.5%						
Left	3	6.5%						
SWE side	46							
Right	20	43.5%						
Left	26	56.5%						
SWE, m/s	46							
Rest	46	3.47	0.78	2.31	2.98	3.31	3.71	6.36
Contraction	46	7.47	1.03	4.81	7.11	7.63	8.08	9.70
Strength, kg	46	6.81	2.75	2.07	4.98	6.13	7.90	13.59

Abbreviations: Std. dev., standard deviation; Pctl., percentile; BMI, body mass index; SWE, shear wave elastography.

**Table 2 jcm-14-01121-t002:** Descriptive statistics by sex (n = 46).

Variable	Male				Female			
	N	Mean	SD	Median	N	Mean	SD	Median
Age, years	14	47.29	15.14	47.5	32	51.72	12.21	53.5
Height, cm	14	173.06	7.93	172.5	32	163.22	7.89	161
Weight, kg	14	76.66	15.52	72.9	32	67.95	13.94	67.5
BMI, kg/m^2^	14	25.58	4.75	23.62	32	25.39	4.25	25.07
Education	14				32			
Primary school	3	21.4%			5	15.6%		
Secondary school	3	21.4%			7	21.9%		
University	8	57.1%			20	62.5%		
Dominant side	14				32			
Right	13	92.9%			30	93.8%		
Left	1	7.1%			2	6.2%		
SWE side	14				32			
Right	7	50%			13	40.6%		
Left	7	50%			19	59.4%		
SWE, m/s								
Rest	14	3.72	0.86	3.52	32	3.36	0.73	3.26
Contraction	14	7.60	1.08	7.847	32	7.42	1.02	7.38
Strength, kg	14	9.54	2.96	9.857	32	5.61	1.58	5.56

Abbreviations: SD, standard deviation; BMI, body mass index; SWE, shear wave elastography.

**Table 3 jcm-14-01121-t003:** Normative values of shear wave elastography at rest (m/s) by age and sex.

Age	Male			Female		
	Pctl. 25	Median	Pctl. 75	Pctl. 25	Median	Pctl. 75
25	2.64	3.06	3.37	2.59	2.96	3.26
26	2.67	3.1	3.38	2.59	2.97	3.29
27	2.73	3.13	3.39	2.6	2.98	3.3
28	2.79	3.15	3.4	2.61	2.99	3.31
29	2.82	3.17	3.42	2.62	3	3.33
30	2.85	3.18	3.45	2.63	3.01	3.34
31	2.87	3.19	3.51	2.64	3.04	3.36
32	2.89	3.2	3.55	2.65	3.06	3.37
33	2.9	3.2	3.57	2.65	3.09	3.37
34	2.91	3.21	3.57	2.68	3.1	3.38
35	2.94	3.23	3.6	2.71	3.12	3.38
36	2.96	3.27	3.65	2.74	3.13	3.39
37	2.98	3.31	3.68	2.77	3.14	3.39
38	3	3.33	3.7	2.8	3.15	3.4
39	3.04	3.36	3.71	2.81	3.16	3.41
40	3.09	3.38	3.73	2.82	3.17	3.42
41	3.13	3.39	3.77	2.83	3.18	3.44
42	3.14	3.4	3.79	2.85	3.18	3.45
43	3.17	3.42	3.8	2.86	3.19	3.48
44	3.18	3.44	3.84	2.87	3.19	3.51
45	3.19	3.49	3.9	2.88	3.19	3.54
46	3.19	3.53	3.98	2.89	3.2	3.55
47	3.2	3.56	4.06	2.9	3.2	3.56
48	3.21	3.57	4.13	2.9	3.2	3.57
49	3.22	3.59	4.19	2.91	3.21	3.57
50	3.25	3.63	4.24	2.92	3.21	3.57
51	3.31	3.67	4.28	2.93	3.21	3.58
52	3.33	3.69	4.31	2.94	3.23	3.6
53	3.35	3.7	4.35	2.96	3.25	3.63
54	3.38	3.73	4.38	2.96	3.27	3.65
55	3.38	3.75	4.42	2.97	3.29	3.66
56	3.39	3.78	4.45	2.98	3.31	3.68
57	3.41	3.8	4.56	2.99	3.32	3.69
58	3.44	3.84	4.66	3	3.34	3.7
59	3.47	3.88	4.77	3.02	3.35	3.7
60	3.52	3.94	4.85	3.05	3.36	3.71
61	3.56	4.02	4.87	3.07	3.37	3.72
62	3.57	4.09	4.89	3.09	3.38	3.73
63	3.58	4.16	4.92	3.12	3.38	3.75
64	3.61	4.22	4.94	3.13	3.39	3.77
65	3.66	4.26	4.95	3.14	3.39	3.78
66	3.69	4.3	4.96	3.15	3.4	3.79
67	3.7	4.33	4.96	3.16	3.41	3.79
68	3.72	4.37	4.96	3.17	3.42	3.8
69	3.74	4.4	4.97	3.18	3.43	3.82
70	3.77	4.43	4.97	3.18	3.44	3.85

Data are presented in m/s. Abbreviations: Pctl., percentile.

**Table 4 jcm-14-01121-t004:** Normative values of shear wave elastography during contraction (m/s) by age and sex.

Age	Male			Female		
	Pctl. 25	Median	Pctl. 75	Pctl. 25	Median	Pctl. 75
25	7.44	8.01	8.51	7.24	7.84	8.31
26	7.43	8	8.5	7.23	7.82	8.3
27	7.43	8	8.5	7.23	7.79	8.29
28	7.41	7.99	8.49	7.22	7.78	8.28
29	7.4	7.99	8.49	7.22	7.76	8.26
30	7.39	7.98	8.49	7.2	7.74	8.25
31	7.37	7.98	8.48	7.19	7.73	8.23
32	7.36	7.97	8.48	7.18	7.72	8.2
33	7.35	7.97	8.47	7.17	7.7	8.17
34	7.34	7.96	8.46	7.16	7.69	8.15
35	7.32	7.95	8.45	7.16	7.69	8.12
36	7.31	7.95	8.44	7.15	7.69	8.12
37	7.3	7.94	8.42	7.15	7.68	8.11
38	7.29	7.93	8.41	7.14	7.68	8.1
39	7.28	7.92	8.4	7.14	7.67	8.09
40	7.26	7.91	8.39	7.13	7.66	8.09
41	7.26	7.89	8.37	7.13	7.66	8.09
42	7.25	7.88	8.36	7.12	7.65	8.08
43	7.25	7.87	8.35	7.12	7.64	8.08
44	7.24	7.86	8.33	7.11	7.63	8.08
45	7.24	7.85	8.32	7.08	7.61	8.08
46	7.23	7.84	8.31	7.06	7.58	8.07
47	7.23	7.83	8.31	7.03	7.56	8.07
48	7.23	7.81	8.3	7	7.53	8.07
49	7.23	7.8	8.29	6.98	7.51	8.06
50	7.23	7.79	8.29	6.95	7.48	8.05
51	7.22	7.78	8.28	6.92	7.46	8.04
52	7.22	7.77	8.27	6.9	7.45	8.03
53	7.22	7.76	8.27	6.87	7.44	8.02
54	7.21	7.76	8.26	6.83	7.43	8.01
55	7.21	7.75	8.26	6.79	7.41	7.99
56	7.2	7.74	8.24	6.76	7.39	7.98
57	7.19	7.73	8.23	6.72	7.37	7.97
58	7.19	7.73	8.22	6.7	7.35	7.96
59	7.18	7.72	8.21	6.69	7.32	7.95
60	7.17	7.71	8.19	6.67	7.3	7.94
61	7.17	7.71	8.18	6.66	7.28	7.92
62	7.17	7.7	8.17	6.65	7.26	7.9
63	7.17	7.69	8.15	6.63	7.25	7.87
64	7.16	7.69	8.14	6.62	7.24	7.85
65	7.16	7.69	8.12	6.61	7.23	7.82
66	7.16	7.69	8.12	6.61	7.23	7.79
67	7.15	7.69	8.12	6.6	7.22	7.78
68	7.15	7.69	8.11	6.59	7.22	7.77
69	7.15	7.68	8.11	6.56	7.21	7.75
70	7.15	7.68	8.11	6.52	7.2	7.74

Data are presented in m/s. Abbreviations: Pctl., percentile.

**Table 5 jcm-14-01121-t005:** Model coefficients for the ordinal cumulative probability models for shear wave elastography measurements.

Variable	Coef. (95% CI)	SE	Wald Z	*p*-Value
Shear wave elastography at rest				
Age	0.08 (0.01 to 0.14)	0.03	2.41	0.02
Sex (=Female)	0.72 (−3.24 to 4.68)	2.02	0.36	0.72
Interaction	0.01 (−0.12 to 0.04)	0.04	−0.97	0.33
Shear wave elastography at rest				
Age	−0.01 (−0.08 to 0.05)	0.03	−0.47	0.64
Sex (=Female)	0.04 (−4.05 to 4.13)	2.09	0.02	0.98
Interaction	−0.01 (−0.09 to 0.07)	0.04	−0.34	0.73

Abbreviations: Coef., unstandardized regression coefficient; CI, confidence interval; SE, standard error.

**Table 6 jcm-14-01121-t006:** Model coefficients for the relationship between shear wave elastography and strength measurements.

Variable	Coef. (95% CI)	SE	t	*p*-Value
Intercept	5.13 (−2.83 to 13.11)	3.95	1.30	0.20
SWE at rest	0.06 (−1.04 to 1.16)	0.55	0.11	0.91
SWE during contraction	0.19 (−0.64 to 1.03)	0.42	0.47	0.64

Abbreviations: SWE, shear wave elastography; Coef., unstandardized regression coefficient; CI, confidence interval; SE, standard error.

## Data Availability

The raw data supporting the conclusions of this article will be made available by the authors on request.

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
