# Peer review of "Normative Data of Supraspinatus Muscle Shear Wave Elastography in Healthy Shoulders: A Cross-Sectional Study"

_jcm, 2025, doi:10.3390/jcm14041121_

Round 1
Reviewer 1 Report
Comments and Suggestions for Authors
Authors present a crossectional study on normative data of supraspinatus muscle shear wave elastography in healthy shoulders to evaluate the relationship between SWE measures and isometric strength, with a a significant increase in muscle stiffness from rest to contraction and a relationship between age and SWE at rest , no differences between male and female. I suggest to include clinical applications of these findings, otherwise the study is very nicely written.
Author Response
Comment 1: Authors present a cross-sectional study on normative data of supraspinatus muscle shear wave elastography in healthy shoulders to evaluate the relationship between SWE measures and isometric strength, with a significant increase in muscle stiffness from rest to contraction and a relationship between age and SWE at rest, no differences between male and female. I suggest including clinical applications of these findings, otherwise the study is very nicely written.
Response 1:
Dear reviewer, thank you for your comment. We have added a new paragraph in the discussion section explaining the possible clinical applications of the results of our study. It reads:
“In summary, the results of the present study can be used to discern if a patient with shoulder symptoms is within the range of values of healthy subjects, implying his muscle is within a normal range of mechanical properties. A subject who is well above or below the normative percentiles could have an underlying pathology (e.g., rotator cuff tendinopathy instead of subacromial bursitis) or be at higher risk of developing a future rotator cuff disorder (e.g., rotator cuff tear). Additionally, a subject outside the normative range limits may also respond poorly to a specific treatment (such as rotator cuff repair). Moreover, these data can be used to assess the patient's progress after surgery or, for example, an exercise program to determine whether the muscle is responding adequately to the intervention.”
Reviewer 2 Report
Comments and Suggestions for Authors
This cross-sectional study by Perez-Porta et al. establishes normative values for shear wave elastography (SWE) measurements of the supraspinatus muscle at rest and during contraction in healthy individuals. Data were collected from 46 participants, and normative values were stratified by age and sex using advanced statistical modeling. The study found that SWE values increased with age at rest but not during contraction, and no significant correlation was observed between SWE measurements and isometric strength. The findings aim to provide reference data for clinical evaluation of shoulder disorders.
Introduction
While the introduction emphasizes the need for normative SWE values, it does not sufficiently explain how these data would directly impact clinical decision-making or improve patient outcomes.
The rationale for focusing solely on the supraspinatus muscle and excluding other shoulder muscles commonly involved in disorders is not adequately addressed.
Results
Although normative values are presented, their practical relevance for diagnosing or managing shoulder disorders is not discussed, leaving a gap in the results' applicability.
Figures showing normative values lack clear annotations to help readers understand trends or differences across age and sex groups.
The study reports variability in SWE measurements but does not analyze potential factors (e.g., BMI, physical activity) that could contribute to this variability.
The results focus heavily on p-values without adequately discussing the clinical implications of observed differences or relationships.
Methods
The limited number of participants, especially in subgroups (e.g., by age and sex), restricts the generalizability of the normative values and may reduce the robustness of the statistical models.
The study does not validate its findings against other populations or datasets, raising concerns about the applicability of the normative values to diverse clinical settings.
While SWE protocols are described, potential sources of measurement error (e.g., probe placement, inter-rater variability) are not adequately addressed.
Discussion
The discussion suggests broad clinical applicability of the normative values without fully addressing the study's limitations, such as the small sample size and single-center design.
The study does not explore how these SWE values could be combined with other diagnostic tools or clinical assessments to improve the evaluation of shoulder disorders.
While the need for further research is acknowledged, specific recommendations for study design, such as larger sample sizes or inclusion of pathological conditions, are not provided.
Author Response
This cross-sectional study by Perez-Porta et al. establishes normative values for shear wave elastography (SWE) measurements of the supraspinatus muscle at rest and during contraction in healthy individuals. Data were collected from 46 participants, and normative values were stratified by age and sex using advanced statistical modeling. The study found that SWE values increased with age at rest but not during contraction, and no significant correlation was observed between SWE measurements and isometric strength. The findings aim to provide reference data for clinical evaluation of shoulder disorders.
Introduction
While the introduction emphasizes the need for normative SWE values, it does not sufficiently explain how these data would directly impact clinical decision-making or improve patient outcomes.
Dear reviewer, thank you for your comment. We have added a paragraph within the introduction section explaining the benefits of obtaining proper normative data of shear wave elastography measures. It reads:
“This information is crucial because it facilitates interpretation of SWE measures of patients with shoulder disorders in clinical practice, that could help to improve diagnosis of shoulder disorders, enhance the monitoring of clinical progression of patients, facilitate the implementation of individualized treatment selection, and allow to early detection of muscle tissues’ alterations in aim to improve risk prediction to avoid possible musculoskeletal injuries.”
The rationale for focusing solely on the supraspinatus muscle and excluding other shoulder muscles commonly involved in disorders is not adequately addressed.
Dear reviewer, we have added a more detailed explanation about the rationale for assessing supraspinatus muscle in the introduction section. It reads:
“Among the rotator cuff muscles, the supraspinatus is the most exposed to mechanical stress due to its role as an abductor, flexor, internal rotator, external rotator, and stabilizer [2]. This situation makes the musculotendinous complex of the supraspinatus the most involved in rotator cuff injuries, both traumatic and atraumatic, making it a good reference point for obtaining values that reflect the overall functional state of the rotator cuff.[6,7]”
Results
Although normative values are presented, their practical relevance for diagnosing or managing shoulder disorders is not discussed, leaving a gap in the results' applicability.
Dear reviewer, thank you for your comment. We have added a new paragraph in the discussion section explaining the possible clinical applications of the results of our study. It reads:
“In summary, the results of the present study can be used to discern if a patient with shoulder symptoms is within the range of values of healthy subjects, implying his muscle is within a normal range of mechanical properties. A subject who is well above or below the normative percentiles could have an underlying pathology (e.g., rotator cuff tendinopathy instead of subacromial bursitis) or be at higher risk of developing a future rotator cuff disorder (e.g., rotator cuff tear). Additionally, a subject outside the normative range limits may also respond poorly to a specific treatment (such as rotator cuff repair). Moreover, these data can be used to assess the patient's progress after surgery or, for example, an exercise program to determine whether the muscle is responding adequately to the intervention.”
Figures showing normative values lack clear annotations to help readers understand trends or differences across age and sex groups.
Dear reviewer, we have added a sentence about “data interpretation” in the footnote of Figures 1 and 2, in aim to help readers understand trends and differences between sex groups. It reads:
Figure 1: “Data interpretation: There is an estimated increase of SWE measures at rest with increasing age, and males show greater SWE values than females, especially in older ages.”
Figure 2: “There is an estimated decrease of SWE measures during contraction with increasing age, and males show greater SWE values than females, which do not appear to be influenced by age.”
The study reports variability in SWE measurements but does not analyze potential factors (e.g., BMI, physical activity) that could contribute to this variability.
Dear reviewer, thank you for pointing out this issue. Due to limitations in the available sample size, we could not include more covariables within the regression models to evaluate other possible causes of variability in SWE measurements. We have added a sentence in the limitations section (at the end of the discussion section) about this issue. It now reads:
“Furthermore, the small sample size limited the number of possible covariates to include in the regression models, so we could not evaluate the influence of other factors such as body mass index or physical activity levels.”
The results focus heavily on p-values without adequately discussing the clinical implications of observed differences or relationships.
Dear reviewer, thank you for your comment. We have carefully reviewed all the manuscript to make sure we don’t focus on p-values, and to ensure we reported in all sections of the manuscript the confidence intervals instead. In the discussion section we have just included confidence intervals of differences and regression coefficients, without naming p-values to avoid misinterpretation based on them. Furthermore, in the discussion section we have modified the sentence in which we talk about the relationship between age and SWE, making it clear that it was a small relationship despite being statistically significant. It now reads:
“On the other hand, there was a significant small positive relationship between age and SWE at rest based on the ordinal cumulative probability models (coefficient, 0.08; 95% CI, 0.01 to 0.14),…”
Methods
The limited number of participants, especially in subgroups (e.g., by age and sex), restricts the generalizability of the normative values and may reduce the robustness of the statistical models.
Dear reviewer, you are right with this comment. We have modified the conclusions section of the study to point out this issue when interpreting results of our investigation. It now reads:
“This cross-sectional study has established normative median, 25th and 75th percentile values of SWE measurements in the supraspinatus muscle at rest and during contraction, by age and sex. However, results should be interpreted cautiously due to the limited sample size, which restricts generalizability and may reduce robustness of the obtained values.”
The study does not validate its findings against other populations or datasets, raising concerns about the applicability of the normative values to diverse clinical settings.
Dear reviewer, we have included this limitation in the limitations section of the manuscript at the end of the discussion section, so future researchers can take it into account. It now reads:
“Finally, our study was conducted in a single center, so our findings should be validated against other populations or datasets to improve applicability to diverse clinical settings.”
While SWE protocols are described, potential sources of measurement error (e.g., probe placement, inter-rater variability) are not adequately addressed.
Dear reviewer, thank you for this comment. Regarding inter-rater variability, all measurements were taken by the same radiologist. One of the radiologist took the SWE measurements, and the other one just helped to positioning the patient and secure the hand-held dynamometer while conducting the contraction measurements. We have modified the methods section to be clear about this point. It now reads:
“One of the radiologists took the SWE measurements, while the other one helped into positioning the patient and securing the hand-held dynamometer.”
On the other hand, regarding the probe positioning, we agree that its variability can influence the measurement error. We followed published protocols for the evaluation of the supraspinatus muscle belly, and the measurements were taken by a radiologist specialized in ultrasound measurements. The possible influence of probe positioning influence the reliability estimates of SWE measurements, meaning that if reliability analyses showed good values, the variability in probe positioning was small, with no great influence on measurement error, which has been shown to be small in the present study.
Discussion
The discussion suggests broad clinical applicability of normative values without fully addressing the study's limitations, such as the small sample size and single-center design.
Dear reviewer, we have modified the limitation section accordingly. It now reads:
“Furthermore, the small sample size limited the number of possible covariates to include in the regression models, so we could not evaluate the influence of other factors such as body mass index or physical activity levels. Finally, our study was conducted in a single center, so our findings should be validated against other populations or datasets to improve applicability to diverse clinical settings.”
The study does not explore how these SWE values could be combined with other diagnostic tools or clinical assessments to improve the evaluation of shoulder disorders.
Dear reviewer, since this is the first conducted study into this topic using proper statistical analysis, our aim was not to evaluate how the combination of SWE measurements with other diagnostic tools or clinical assessments could improve the evaluation of shoulder disorders. We have added this topic into the “future studies recommendations” section so other researchers can take it into account for future investigations. It now reads:
“Future studies should be conducted with greater sample sizes to externally vali-date our range of normative values and to allow the possibility of including more co-variates (e.g., body mass index or physical activity level) and their interactions. Furthermore, future studies should be conducted using multi-center design and including subjects with shoulder pathological conditions to improve robustness and applicability of the results. Furthermore, future studies should also evaluate how these SWE measurements combine with other diagnostic tools and/or clinical assessments to improve the evaluation and management of shoulder disorders.”
While the need for further research is acknowledged, specific recommendations for study design, such as larger sample sizes or inclusion of pathological conditions, are not provided.
Dear reviewer, we have modified the “future studies recommendations” section at the end of the discussion section accordingly. It now reads:
“Future studies should be conducted with greater sample sizes to externally vali-date our range of normative values and to allow the possibility of including more co-variates (e.g., body mass index or physical activity level) and their interactions. Furthermore, future studies should be conducted using multi-center design and including subjects with shoulder pathological conditions to improve robustness and applicability of the results. Furthermore, future studies should also evaluate how these SWE measurements combine with other diagnostic tools and/or clinical assessments to improve the evaluation and management of shoulder disorders.”
Round 2
Reviewer 2 Report
Comments and Suggestions for Authors
All my concerns have been addressed.